# Visible-light-driven amino acids production from biomass-based feedstocks over ultrathin CdS nanosheets

Song Song[1,4], Jiafu Qu[2,4], Peijie Han[3,4], Max J. Hülsey [1], Guping Zhang[2], Yunzhu Wang [1], Shuai Wang [3✉], Dongyun Chen[2], Jianmei Lu[2✉] & Ning Yan [1✉]

Chemical synthesis of amino acids from renewable sources is an alternative route to the current processes based on fermentation. Here, we report visible-light-driven amination of biomass-derived α-hydroxyl acids and glucose into amino acids using $NH_3$ at 50 °C. Ultrathin CdS nanosheets are identified as an efficient and stable catalyst, exhibiting an order of magnitude higher activity towards alanine production from lactic acid compared to commercial CdS as well as CdS nanoobjects bearing other morphologies. Its unique catalytic property is attributed mainly to the preferential formation of oxygen-centered radicals to promote α-hydroxyl acids conversion to α-keto acids, and partially to the poor $H_2$ evolution which is an undesired side reaction. Encouragingly, a number of amino acids are prepared using the current protocol, and one-pot photocatalytic conversion of glucose to alanine is also achieved. This work offers an effective catalytic system for amino acid synthesis from biomass feedstocks under mild conditions.

[1] Department of Chemical and Biomolecular Engineering, National University of Singapore, 4 Engineering Drive 4, Singapore 117585, Singapore. [2] College of Chemistry, Chemical Engineering and Materials Science, Collaborative Innovation Center of Suzhou Nano Science and Technology, Soochow University, 215123 Suzhou, China. [3] State Key Laboratory for Physical Chemistry of Solid Surfaces, Collaborative Innovation Center of Chemistry for Energy Materials, National Engineering Laboratory for Green Chemical Productions of Alcohols-Ethers-Esters, and College of Chemistry and Chemical Engineering, Xiamen University, 361005 Xiamen, China. [4] These authors contributed equally: Song Song, Jiafu Qu, Peijie Han. ✉email: shuaiwang@xmu.edu.cn; lujm@suda.edu.cn; ning.yan@nus.edu.sg

Amino acids are the building blocks of life, and are widely used in nutrition, agriculture, polymer synthesis, and pharmaceutical industry[1–3]. Nowadays, amino acids are mainly produced via fermentation processes, which require long reaction time, involve complex purification steps, generate a large number of inorganic salts, and are energy-intensive due to heavy use of compressor[4,5]. Another limitation of the biological process is that only amino acids in L-configuration can be produced, but D-amino acids are finding increasing applications and are normally much more valuable. Chemical synthesis of amino acids could potentially avoid many limitations mentioned above, and it produces a racemic mixture, which, after optical resolution, affords amino acids in both D- and L-configurations. Several methods such as Strecker synthesis[6] have long been established, however, highly toxic cyanides and non-renewable aldehydes are used as substrates preventing their large-scale application[7,8]. The chemical routes to transform renewable organic carbon feedstocks and cheap, abundant $NH_3$ into amino acids[9,10], and other relevant nitrogen-containing chemicals[11–16] in an energy-saving manner are highly desirable. For example, an exciting progress has recently been reported on the sustainable synthesis of glycine and alanine by catalytic fixation of $N_2$ and $CO_2$[17].

Chemical conversion of woody biomass components into α-amino acids is non-trivial. Despite various efforts to convert lignocellulose components into various chemicals including a number of organic acids[18–22], chemical transformation of biomass into amino acids is rare. Previously, we proposed a two-step protocol to firstly convert biomass components into α-hydroxyl acids via either biological[23,24] or chemical processes[19,25–31], followed by thermocatalytic amination to convert the -OH groups into the -NH₂ groups[9,32,33]. Several limitations remain for the amination reaction: (1) it proceeds under relatively harsh reaction conditions (220 °C); (2) it requires the use of hydrogen gas (10 bar), and (3) only noble metal catalysts are effective[32]. These drawbacks undermine the comparative advantages of the reported chemical routes compared to the existing fermentation processes. Indeed, amination of -OH groups of alcohols, α-hydroxyl acids, or their derivatives into corresponding amines using ammonia or organic nitrogen sources under mild reaction conditions (<50 °C in the absence of $H_2$) remains a grand challenge over both homogeneous and heterogeneous catalytic systems[34–37].

Photocatalysis utilizes photogenerated electrons and holes to activate substrates, thereby avoiding high temperature and/or high pressure operation[38–44]. As such, photocatalysis is particularly suitable for biomass conversion, since the selectivity of desired products from highly functionalized biomass or bio-based feedstock is often compromised under severe thermal-catalytic conditions. Indeed, photocatalysis has recently shown encouraging potentials in the valorization of biomass feedstocks[45]. Wang and co-workers[46] reported the photocatalytic coproduction of diesel precursors and hydrogen from biomass-derived furans using Ru-doped $ZnIn_2S_4$ catalysts, alkane formation from fatty acid over $Pt/TiO_2$ photocatalysts[47], as well as the production of methanol and syngas from bio-polyols and sugars[48]. Reisner and co-workers[49] studied the photocatalytic conversion of lignocellulose into $H_2$ over cadmium sulfide quantum dots. Wang and co-workers[50] reported fractionation and conversion of native lignin under light irradiation condition. Oxidation of biomass-derived 5-hydroxymethylfurfural (HMF) to 2,5-furandicarboxylic acid (FDCA) with Co-thioporphyrazine bonded $g$-$C_3N_4$[51] or Ni/CdS catalyst have also been reported[52]. Despite these advances, limited studies have been performed to make high-value chemicals from biomass. For photocatalytic amino acid synthesis, only commercially available semiconductors in the bulk form were tested and the activity was too low to be of synthetic interest[53,54]. And, to our knowledge, there has been no progress for such transformation in the past two decades.

Herein, we present visible-light-driven, CdS nanosheet catalyzed formation of amino acids from biomass-derived α-hydroxyl acids and even directly from sugars at 50 °C and 1 bar $N_2$ (Fig. 1). A broad range of biomass-derived hydroxyl acids is applied for the photocatalytic synthesis of different amino acids. Morphology plays a key role in the activity and determines whether the selectivity is directed towards amination or undesired water splitting reaction. The present work highlights the significance of morphology control in designing photocatalysts for biomass valorization through tuning the specific activity and selectivity of semiconductor photocatalysts.

## Results

**Photocatalytic amination of lactic acid to alanine.** We initially conducted experiments over several typical semiconductors ($TiO_2$, $g$-$C_3N_4$, ZnO, $BiVO_4$, CuS, and CdS) using lactic acid as a model compound (Fig. 2a and Supplementary Fig. 1). Lactic acid is a readily available feedstock from glucose and cellulose[55]. Under UV-vis light irradiation, $TiO_2$ showed the best activity (0.4 mmol $g_{cat}^{-1} h^{-1}$) among the selected photocatalysts ($TiO_2$, ZnO, and $g$-$C_3N_4$). However, no activity was observed over $TiO_2$ under visible-light irradiation (420–780 nm) due to its large band gap energy (3.2 eV, $\lambda = 387$ nm). In contrast, commercial CdS showed an activity of 1.3 mmol $g_{cat}^{-1} h^{-1}$ in the amination of lactic acid to alanine under the same condition. Although the

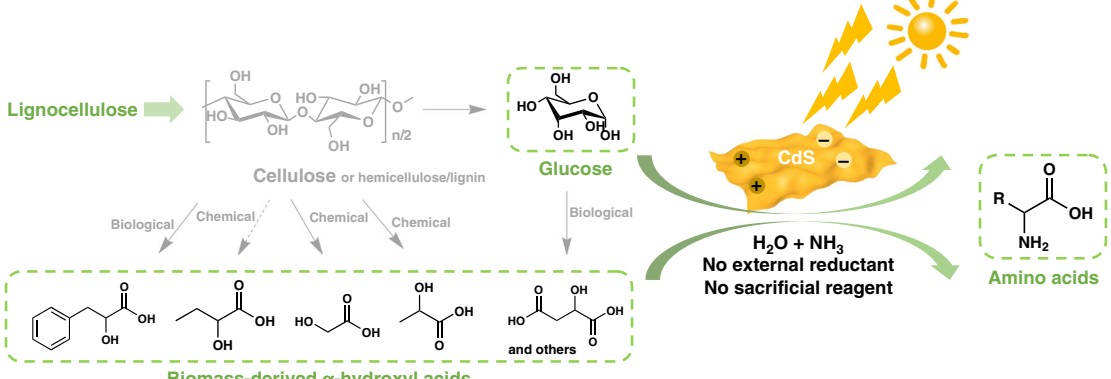

**Fig. 1 Photocatalytic amination of glucose or biomass-derived α-hydroxyl acids to amino acids.** Woody biomass components, including cellulose, hemicellulose, and lignin, could be converted into α-hydroxyl acids (left). In this work, these acids has been photo-catalytically converted into several important amino acids under visible-light irradiation over CdS nanosheets (right). Glucose is also converted into alanine in a single-step.

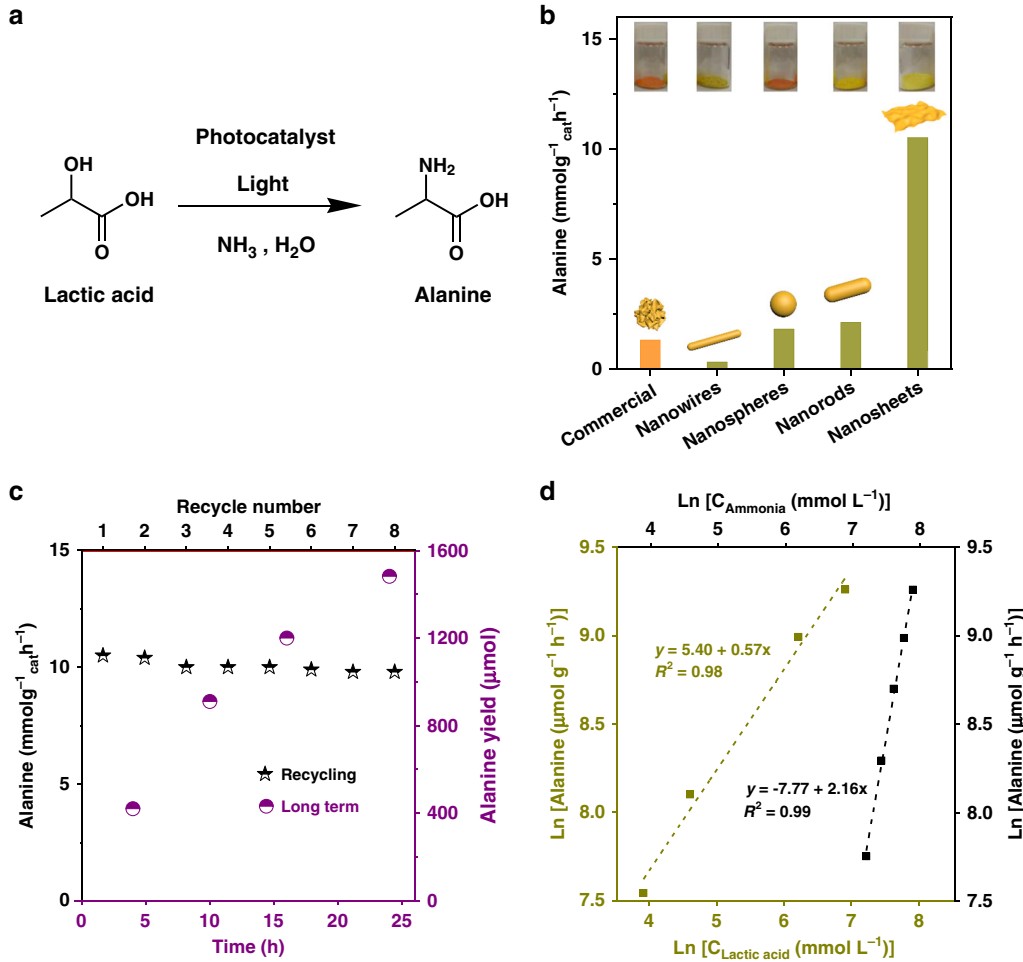

**Fig. 2 Photocatalytic synthesis of alanine from biomass-derived lactic acid. a** Scheme for amination of lactic acid to alanine with ammonia. **b** Alanine production rate over CdS catalysts with different types of morphology under visible light. **c** Recycling study and stability over CdS nanosheets. Reaction conditions: 10 mg photocatalyst, 20 mmol lactic acid, 4 mL ammonia solution (25 wt%), 16 mL deaerated water, $T = 50$ °C, and 1 bar $N_2$. For recycle experiments, $t = 4$ h. The black star in **c** denotes recycling activity. The purple cycle in **c** denotes long-term activity. **d** Logarithm of rates versus logarithm of lactic acid and ammonia concentrations. Reaction conditions: 10 mg nanosheets, 20 mL solution (lactic acid + ammonia + $H_2O$), $T = 50$ °C, 1 bar $N_2$, visible light, and $t = 4$ h. The yellow dot line and square in **d** denote lactic acid. The black dot line and square in **d** denote ammonia.

activity was low, CdS represents the only material able to promote the desired transformation under visible light and was selected for further engineering to enhance its activity.

It has been well established in $H_2$ production and C–H bond oxidation reactions that the morphology of CdS photocatalysts strongly influences the photogenerated electron-hole transfer, thus affecting the photocatalytic performance[56,57]. Inspired by these reports, we prepared CdS bearing four different morphologies, including nanowires, nanospheres, nanorods, and nanosheets and applied them in photo-amination of lactic acid to alanine (Fig. 2b). A striking morphology-dependant activity of CdS was observed. With nanowires as catalyst, alanine formation rate was only 0.3 mmol $g_{cat}^{-1} h^{-1}$. CdS nanospheres and nanorods exhibited much higher activity than CdS nanowires, reaching 1.8 and 2.1 mmol $g_{cat}^{-1} h^{-1}$ production rate, respectively. This value is slightly higher than that of the commercial CdS. Remarkably, the alanine formation rate reached 10.5 mmol $g_{cat}^{-1} h^{-1}$ over CdS nanosheets, an order of magnitude higher than over commercial CdS, CdS nanospheres, and CdS nanorods, and 2 orders of magnitude higher than CdS nanowires and commercial $TiO_2$ (under UV-vis light irradiation).

Encouraged by the extraordinary activity of CdS nanosheets, a long-term irradiation experiment was also conducted. The results

were shown in Fig. 2c. In all, 420 μmol alanine was produced after 4 h irradiation. With prolonging reaction time to 10 h and 24 h, the yield of alanine further increased to 910 μmol and 1482 μmol, respectively. Over the entire process, an almost linear increase of product over time was observed. After 24 h reaction, ICP analysis of the reaction solution after catalyst removal indicted only 5 ppm $Cd^{2+}$. Lactamide, alaninamide, acetic acid, and paraldehyde were detected as the main side products in photocatalytic amination reaction (Supplementary Fig. 2), based on which a reaction network is proposed (Supplementary Fig. 3). A control experiment using alanine as substrate demonstrated negligible further oxidation of alanine back to pyruvic acid (Supplementary Fig. 4). In the recycling experiments, no obvious activity loss was noticed even after eight repeated use of catalyst, again highlighting the exceptional stability of nanosheets. Interestingly, the formation of alanine is half-order with respect to lactic acid, and second order with respect to ammonia (Fig. 2d). The mechanistic assessment shown below indicates that the cleavage of $C_\alpha$-H bond in lactic acid adsorbed on CdS is the kinetically relevant step for photocatalytic amination. Accordingly, the measured apparent reaction order of ~0.5 on lactic acid suggests that the surface of CdS is partially covered by lactic acid at the examined reaction condition, because negligible and saturated coverages would lead

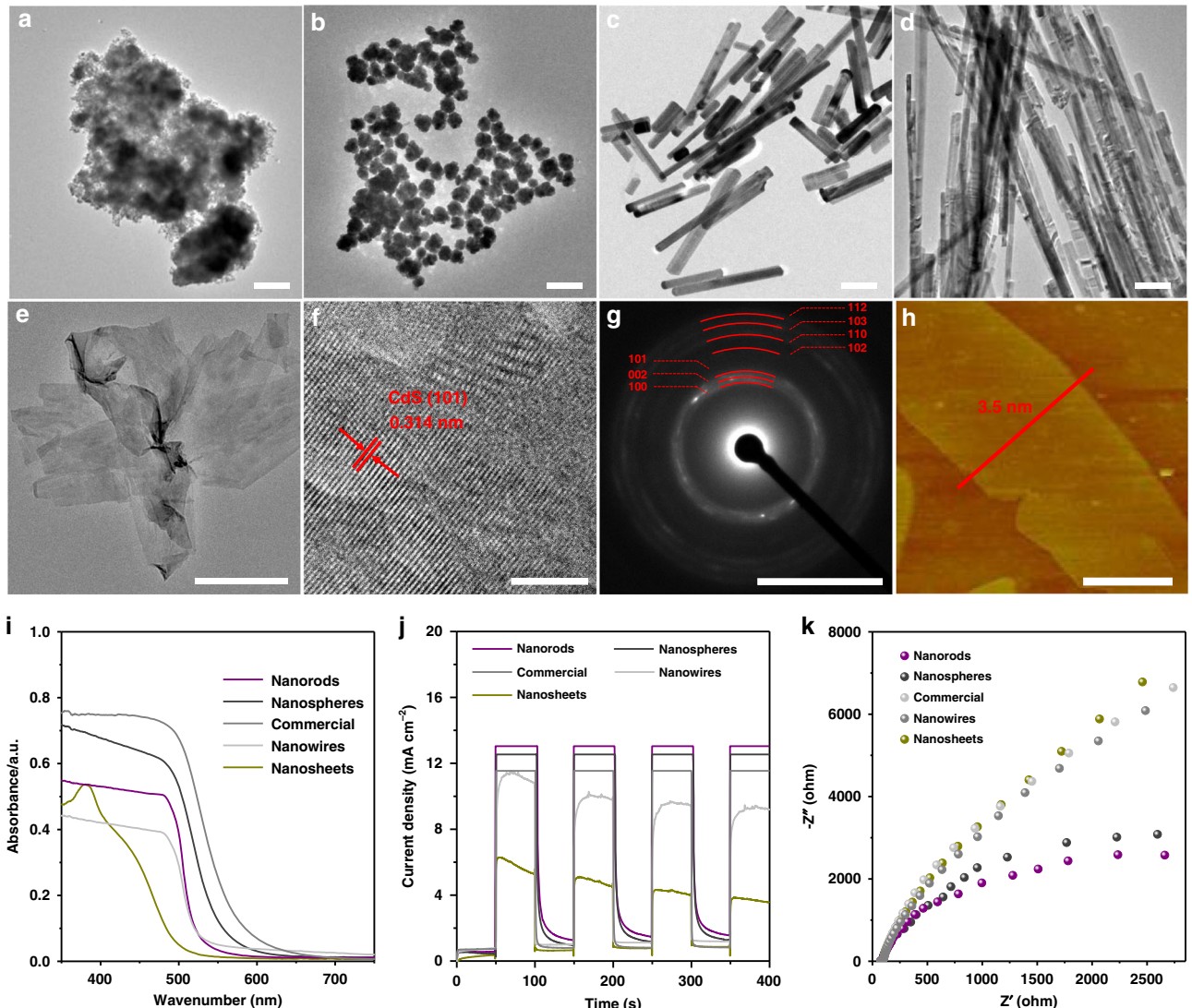

**Fig. 3 Characterization of different morphology CdS.** TEM image of **a** commercial CdS. **b** Nanospheres. **c** Nanorods. **d** Nanowires. **e** Nanosheets. **f** High-resolution TEM of nanosheets. **g** Electron diffraction patterns of nanosheets. **h** AFM images of nanosheets. **i** UV-Vis spectra. **j** Photocurrent transient spectra. **k** Electrochemical impedance spectroscopy (EIS) Nyquist plots. Scale bar: **a–e** 200 nm; **f** 5 nm; **g** 5 nm$^{-1}$; **h** 1 µm.

to first- and zero-order kinetics, respectively. On the other hand, the second order on ammonia implies two ammonia molecules are involved in the $C_\alpha$-H bond abstraction of lactic acid. We surmise that one ammonia interacts with the carboxylic H⁻atom in lactic acid and the other connects to the corresponding α-hydroxyl H⁻atom via H-bonding for stabilizing the transition state of the $C_\alpha$-H cleavage step.

**Catalyst characterization.** The as-synthesized CdS materials were characterized by transmission electron microscope (TEM; Figs. 3a–e). While commercial CdS were heavily aggregated (Fig. 3a), CdS nanospheres were composed of well-distributed particles with diameters ranging from 40 to 60 nm (Fig. 3b). Nanorods and nanowires have similar diameters of ca. 50 nm, and the length of the two are ca. 500 nm and 4 µm, respectively (Figs. 3c, d). The ultrathin nature of CdS nanosheets was confirmed by TEM and atomic force microscope (AFM), with a thickness of ~3.5 nm (Figs. 3e–h). X-ray diffraction (XRD) patterns demonstrated the hexagonal crystal structure in all CdS materials (Supplementary Fig. 5). X-ray photoelectron spectroscopy (XPS) spectra confirmed the presence of elements Cd, S,

and O and the high-resolution spectra of Cd 3$d$, S 2$p$, and O 1$s$ were analyzed (Supplementary Fig. 6). The peak areas of Cd 3$d_{5/2}$ and S 2$p_{3/2}$ were applied for the semi-quantitative elemental analysis of Cd and S. The atomic ratio of Cd and S were close to 1:1 for all CdS materials, which confirms that the samples are pure CdS. The oxygen in CdS possibly resulted from the sample exposure to air. The BET surface area of nanosheets, commercial CdS, nanospheres, nanorods, and nanowires were measured to be 123.4, 58.2, 17.3, 16.4, and 12.1 m$^2$ g$^{-1}$, respectively (Supplementary Fig. 7 and Supplementary Table 1). Considering the surface area difference among various morphology CdS may affect the photocatalytic activity, a set of control experiments were conducted by using different amounts of CdS so that in each experiment the employed CdS catalyst had the same exposed surface area (Supplementary Table 2). The observed activity trend was exactly the same as that shown in Fig. 1b, that is, CdS nanosheets » CdS nanorods > CdS nanospheres > CdS commercial » CdS nanowires. As such, the possibility that the strong morphology-activity correlation is due to the difference of the specific surface area of CdS is excluded.

UV-Vis spectra for different morphology CdS were shown in Fig. 3i. The band gaps for nanosheets, nanorods, nanowires,

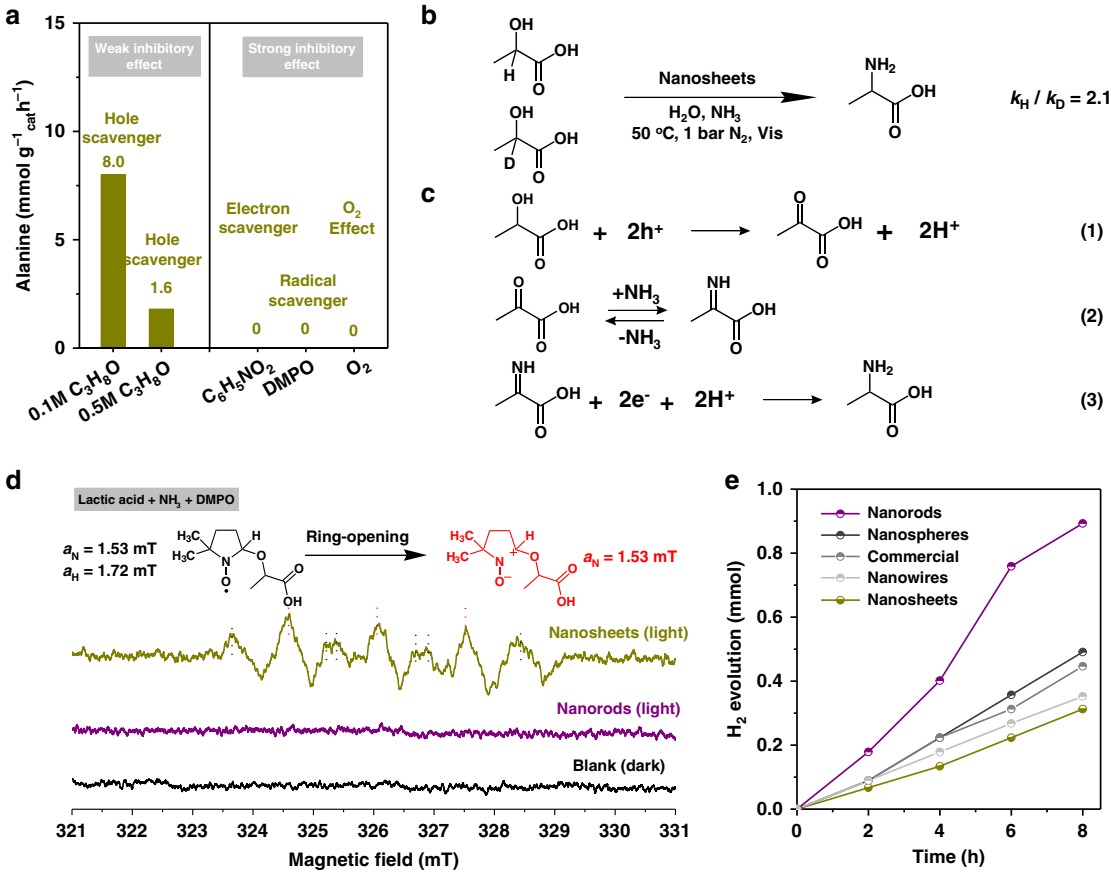

**Fig. 4 Mechanistic insights for photocatalysis amination of lactic acid to alanine. a** Control experiments with different scavengers and gas. Reaction conditions: 10 mg nanosheets, 20 mmol lactic acid, 2 mmol scavenger, 4 mL ammonia solution (25 wt%), 16 mL deaerated water, $T = 50\ ^\circ\text{C}$, 1 bar $N_2$ or $O_2$, and $t = 4$ h. **b** Isotope experiments. Reaction conditions: 10 mg nanosheets, 20 mmol lactic acid or lactic acid-$d_1$, 4 mL ammonia solution (25 wt%), 16 mL deaerated water, $T = 50\ ^\circ\text{C}$, 1 bar $N_2$, and $t = 4$ h. **c** Equations for photocatalysis amination of lactic acid to alanine. **d** In situ ESR spectra for CdS nanosheets and nanorods in a mixture solution of lactic acid and ammonia in the presence of spin-trapping agent 5, 5-dimethyl-1-pyrroline N-oxide (DMPO) with or without visible-light irradiation. **e** $H_2$ evolution experiments. Reaction conditions: 10 mg CdS, 20 mmol lactic acid, 20 mL deaerated water, $T = 50\ ^\circ\text{C}$, and 1 bar $N_2$. $C_6H_5NO_2$ denotes nitrobenzene, and $C_3H_8O$ denotes isopropanol.

nanospheres, and commercial CdS were calculated to be 2.38, 2.33, 2.29, 2.16, and 2.10 eV, respectively[56,58]. Although increasing in band gap could contribute to the enhanced photocatalytic activity through changes in the redox potential for photocatalytic reactions[50,59], no obvious correlation was established between reaction activity and band gap in our system. The photocurrent density spectra and electrochemical impedance spectroscopy (EIS) Nyquist plot were studied to understand the photogenerated charge carrier transfer ability (Fig. 3j, k). Normally, a photocatalyst exhibiting lower photocurrent response and bigger arc radius triggers less effective photogenerated electrons and has poorer $e^-$ $- h^+$ pairs separation ability under light irradiation, which has a detrimental effect on photocatalytic performance[43]. This seems to be the case for CdS nanorods, nanospheres, nanowires, and commercial CdS (Fig. 2b). In sharp contrast, CdS nanosheets had the poorest photoelectrochemical response but exhibited the best catalytic activity toward alanine production. This abnormal phenomenon prompted us to perform further studies to understand the reaction mechanism on photocatalytic amination of lactic acid to alanine and the origin of activity difference among CdS catalysts bearing different morphology.

**Reaction mechanism**. The reaction ceased when nitrobenzene ($C_6H_5NO_2$) was added as an electron scavenger, (Fig. 4a), indicating that the photogenerated electrons are necessary for the

amination of lactic acid to alanine. With 0.1 M and 0.5 M isopropanol ($C_3H_8O$) added as hole scavengers, the formation rate of alanine decreased from 10.5 to 8.0 and 1.6 mmol $g^{-1}h^{-1}$, respectively. Normally, lactic acid itself was used as hole scavenger in many photocatalysis systems to promote hydrogen production, especially in CdS-based systems[60,61]. The weak inhibitory effect of hole scavenger mainly resulted from the hole consumption competition between lactic acid and isopropanol. From the above, it is likely that both photogenerated electrons and holes participated in the formation of alanine from lactic acid. The full utilization of photogenerated electrons and holes also explains the high stability of nanosheets in the absence of any sacrificial reagent. The addition of radical scavenger, 5, 5-dimethyl-1-pyrroline N-oxide (DMPO), totally suppressed the reaction, suggesting the reaction proceeded via radical intermediates. We also found that $O_2$ is detrimental to the photocatalysis system. CdS nanosheet solids disappeared while a transparent reaction mixture was obtained after reaction under $O_2$ due to the decomposition of CdS. Isotope experiments were conducted with $C_\alpha$–D labeled lactic acid as a substrate (Fig. 4b). The extent of kinetic isotope effect (KIE) was calculated based on the yield of alanine. A KIE value of 2.1 was obtained, demonstrating that the C–H abstraction is a kinetically relevant step during the reaction. No $\alpha$-amino isobutyric acid was detected when $\alpha$-hydroxyl isobutyric acid (the hydrogen of $C_\alpha$ was replaced by a methyl group) was used

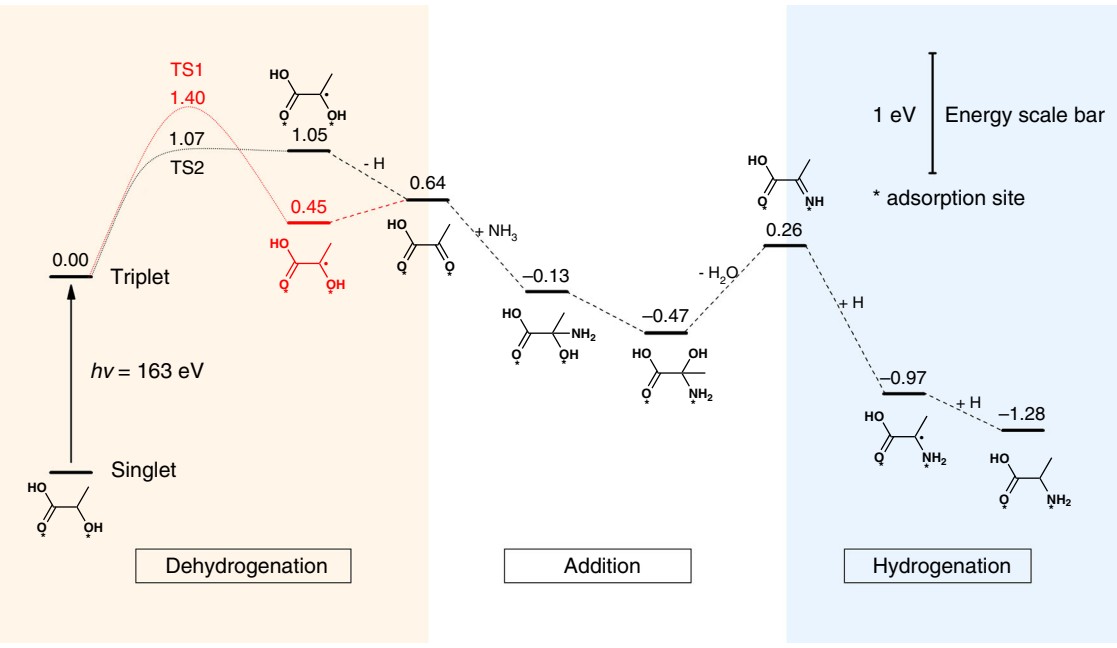

**Fig. 5 Density functional theory (DFT) calculations.** Energy profiles for different reaction steps in lactic acid amination to alanine on CdS (100).

as substrate (Supplementary Fig. 8). Thus, the presence of $C_\alpha$–H is essential for the formation of amino acid.

Pyruvic acid was a key intermediate during the thermocatalytic conversion of lactic acid to alanine, and dehydrogenation of lactic acid to pyruvic acid was identified as the rate-determining step[32]. We conducted control experiments to understand the role of pyruvic acid in the current photocatalysis system. Due to the facile formation of an imine between ammonia and pyruvic acid, a control experiment in the absence of ammonia was conducted using lactic acid and CdS nanosheets under visible-light irradiation. HPLC analysis confirmed the increased formation of pyruvic acid as a function of irradiation time (Supplementary Fig. 9). Further, when pyruvic acid and $NH_3$ were used as substrates, the formation of alanine was observed (Supplementary Table 3). Based on the discussions above, a detailed photocatalytic reaction sequence was depicted (Fig. 4c). Lactic acid is transformed into pyruvic acid by the holes generated on CdS, which further reacts with $NH_3$ to form an imine. Imine accepts protons and electrons to convert into alanine. Interestingly, all CdS materials provided essentially the same formation rate regardless of catalyst morphology when pyruvic acid was used as substrate, highlighting that the rate-determining step in photo-catalytic generation of alanine is the formation of pyruvic acid from lactic acid (step 1 in Fig. 4c), and that CdS nanosheets are exceptional in promoting this step (Supplementary Table 3).

Electron spin resonance (ESR) is a useful technology to detect intermediate radicals during photocatalysis[62]. In situ ESR spectro-scopic studies with DMPO as spin trap agent in the presence of lactic acid and ammonia were conducted (Fig. 4d). The results revealed the formation of oxygen-centered radicals DMPO-OCR ($a_N = 1.53$ mT, $a_H = 1.72$ mT) over CdS nanosheets[63–65]. Another signal ($a_N = 1.5$ mT) belonging to the decomposition of DMPO-OCR by ring-opening was also observed[66]. However, no signals were observed when nanorods and other morphology CdS were used as catalyst. The oxygen-centered radicals (·OCR) resulting from the O–H activation are likely responsible for the formation of pyruvic acid from lactic acid (which was confirmed by DFT study shown in a later section), thus promoting alanine production. On the other hand, band position is not the major reason for the unique activity of CdS nanosheets, as all CdS

materials have similar band positions and are all consistent with that of the redox potential of the key steps in the reaction (Supplementary Figs. 10 and 11 and Supplementary Tables 4 and 5).

As a major side reaction consuming electrons, $H_2$ production over different morphology CdS under visible-light irradiation with lactic acid as hole scavengers were studied (Fig. 4e). CdS nanosheets exhibited the poorest $H_2$ production ability, in accordance with the lowest photocurrent density and largest arc radius in the photoelectrochemical measurements (Fig. 3j, k). Based on the reaction mechanism, the imino acid coming from equilibrium between pyruvic acid and ammonia is reduced by the photogenerated electrons to produce alanine. One can envisage the competitive consumption of electrons in $H_2$ generation is detrimental for imino acid reduction, and therefore the unfavor-able formation of $H_2$ on CdS nanosheets is another reason for its high catalytic activity towards alanine formation.

Density functional theory (DFT) method was applied to elucidate the photocatalytic pathway of lactic acid conversion to amino acid on CdS surfaces. From experimental results, the calculation divided the entire reaction into three steps including lactic acid dehydrogenation, addition of pyruvic acid with ammonia to imino acid, and finally reduction of imino acid to alanine (Fig. 5). In particular, the abstraction of the first H atom from lactic acid in the initial dehydrogenation step appears to be the overall rate-determining step. Here we examined the mechanism of lactic acid dehydrogenation using the hexagonal wurtzite CdS (100) surface, which is the predominant surface exposed on the CdS nanosheet catalyst. The DFT calculations show that lactic acid prefers to adsorb at the CdS (100) surface by coordinating the carbonyl O atom and the α-hydroxy O atom to two vicinal Cd centers (Supplementary Fig. 12). Dehydrogenation of this bound lactic acid species can be initiated by a neighboring S site via cleavage of its O–H bond or the $C_\alpha$–H bond (Supplementary Fig. 12b). The former pathway presents a much lower activation barrier than the latter (1.07 vs. 1.40 eV, with respect to the triplet state of the adsorbed reactant, Fig. 5), benefit from the short distance between the hydroxy H atom and the S site (0.270 nm). This result indicates that lactic acid dehy-drogenation on CdS occurs primarily via dissociation of its

**Table 1 Photocatalytic amination of hydroxyl acids to amino acids over CdS nanosheets[a].**

| Entry | Substrate | Product | $C_{\alpha}$-H or $C_{\beta}$-H bond energy (kcal·mol$^{-1}$) [b] | Formation rate (mmol·g$^{-1}$·h$^{-1}$) | Overall yield (%) |
|-------|-----------|---------|------|------|------|
| 1 |  |  | 88.1 | 7.9 | 31.6 |
| 2 |  |  | 88.3 | 5.4 | 21.6 |
| 3 |  |  | 88.6 | 5.6 | 22.4 |
| 4 |  |  | 88.8 | 5.8 | 23.2 |
| 5 |  |  | 89.6 | 1.4 | 5.6 |
| 6 |  |  | 91.2 | 0.3 | 1.2 |
| 7 |  |  | 91.7 | 0.1 | 0.4 |
| 8 |  |  | 98.0 | - | - |
| 9 |  |  | 76.4 | - | - |

[a] Reaction conditions: substrates 2 mmol, nanosheets 10 mg, 25 wt% ammonia solution 4 mL, 16 mL deaerated water, $T = 50$ °C, $t = 8$ h, 1 bar N$_2$, and visible light.
[b] C–H bond energy was calculated at the B3LYP/def2TZVP level of theory.

α-hydroxy group that yields an oxygen-centered radical, fully consistent with the ESR signals of such radicals captured during photocatalysis (Fig. 4d). The following addition and hydrogenation steps after the lactic acid dehydrogenation were also examined using DFT with optimized structures shown in Supplementary Fig 13, which confirms that lactic acid conversion to amino acid is thermodynamically favorable under light irradiation.

**Other amino acids**. A series of biomass-derived hydroxyl acids were applied to verify the scope of the photocatalytic amination strategy. Leucine, valine, phenylalanine, and α-aminobutyric acid were obtained efficiently from α-hydroxyisocaproic acid, α-hydroxy-3-methylbutyric acid, phenyllactic acid, and α-hydroxybutyric acid over CdS nanosheets, respectively. The formation rate for leucine, valine, phenylalanine, and α-aminobutyric acid reached 7.9, 5.4, 5.6, and 5.8 mmol g$^{-1}$ h$^{-1}$, respectively (Table 1, entries 1, 2, 3, and 4). Aspartic acid (1.4 mmol g$^{-1}$ h$^{-1}$) and glycine (0.3 mmol g$^{-1}$ h$^{-1}$) were obtained by amination of 2-hydroxysuccinic acid and glycolic acid, however, the formation rates were much lower (Table 1, entries 5 and 6). Mandelic acid as an aromatic α-hydroxyl acid was also tested, but no amino acid was detected (Table 1, entry 9). We also explored the production of β-amino acids from β-hydroxyl acids. Trace amounts of 3-aminobutyric acid were detected when 3-hydroxybutyric acid was used as substrates (Table 1, entry 7). No activity was observed when 3-hydroxypropionic acid was applied (Table 1, entry 8). DFT calculations on C$_α$-H or C$_β$-H bond energy of hydroxyl acids were conducted to understand the activity difference of different substrates. In general, an excellent match between C$_α$–H or C$_β$–H bond energy and catalytic activity was observed, i.e., the higher the C–H bond dissociation energy, the lower the catalyst activity. This is in agreement with the earlier proposal that the C–H breakage is the rate-determining step. The only exception is the mandelic acid, for which no amino acid product was identified despite of the very low C$_α$–H bond energy (~76.4 kcal/mol), presumably due to the steric hindrance from the benzene ring. The cyclic voltammograms of hydroxyl acids and related amino acids were conducted (Supplementary Fig. 14). The reversible anodic and cathodic peaks were detected with hydroxyl acids, possibly coming from the redox reactions between hydroxyl acids and related keto acids. The relatively high current density of lactic acid, α-hydroxyisocaproic acid, α-hydroxy-3-methylbutyric acid, phenyllactic acid, and α-hydroxybutyric acid were observed, which may explain their faster amino acid formation rate compared to other substrates. No obvious peak was observed for amino acids except leucine, meaning most amino acids studied in this work are stable.

**Glucose to alanine**. Finally, we tested the possibility of direct photocatalytic conversion of glucose to alanine. A plausible pathway from glucose to alanine via lactic acid as a key intermediate is shown in Supplementary Fig. 15. The glucose concentration was set at 0.1 M, which is a common value used in the literature to produce lactic acid[67]. After 8 h visible-light irradiation, only fructose was detected as the main product over CdS nanosheets, suggesting CdS alone is not able to promote C–C bond cleavage or carboxylate group formation. Consequently, different bases were used as additives due to their ability to promote glucose transformation into lactic acid (Supplementary Table 6)[67]. To our delight, alanine was detected after the addition of different bases, with the highest alanine productivity observed using NaOH. Under the optimized reaction conditions, the productivity of alanine from glucose reached 0.34 mmol g$^{-1}$ h$^{-1}$. The lower productivity of alanine starting from glucose was due to the inhibition effect of OH$^-$ to CdS nanosheets, as in control experiments we found that higher OH$^-$ concentration would significantly suppress the efficiency of photocatalytic amination of lactic acid to alanine (Supplementary Table 7). CdS nanosheets remain much more active than CdS materials bearing other morphologies in converting glucose to alanine (Supplementary Table 6).

## Discussion
In summary, we developed an ultrathin CdS nanosheet catalyst to synthesize amino acids from biomass-derived α-hydroxyl acids

under visible-light irradiation, which exhibited a remarkable activity (10.5 mmol g$^{-1}$ h$^{-1}$, alanine) compared to commercial CdS and CdS with other morphology. Mechanistic studies reveal that the hydroxyl groups in hydroxyl acids are activated to amine groups via an electron-hole coupled mechanism with keto acids as intermediate. In situ ESR experiments and DFT calculations demonstrate the preferential formation of oxygen-centered radical intermediates on the ultrathin CdS nanosheets, which lowered the formation barrier of amino acids leading to high activity. The catalytic system is able to prepare a range of α-amino acids from corresponding hydroxyl acids when the C$_α$–H bond dissociation energy is below 90 kcal mol$^{-1}$. Interestingly, the most effective CdS nanosheet catalyst exhibited a low photocurrent density and a large arc radius, which are typical seen as detrimental to catalytic activity, highlighting common wisdom in photocatalysis may not be applicable to understand or design the photocatalytic conversion of complex bio-based feedstocks into fine chemicals such as amino acids. In nature, herbivores (the major meat source for human beings) consume lignocellulosic materials and transform them into various amino acids followed by the formation of proteins in vivo. This highly delicate biological process has been partially reproduced via photocatalysis in the current study. Identifying base-resistant photocatalysts to enhance the one-pot conversion of sugars into various amino acids are currently underway.

## Methods
**Synthesis of CdS nanosheets**. CdS nanosheets were synthesized according to a modified literature method[68]. Typically, 2.0 mmol Cd(OAc)$_2$·2H$_2$O and 6.0 mmol SC(NH$_2$)$_2$ were dispersed in 60 mL ethylenediamine and then transferred to a Teflon-line stainless-steel autoclave (80 mL), heated at 100 °C for 8 h. After the reaction, the bright yellow product was separated out and washed with deionized water and ethanol several times, and then dried at vacuum oven.

**Synthesis of CdS nanospheres**. CdS nanospheres were synthesized according to a literature method[69]. In all, 0.6 mmol of Cd(OAc)$_2$·2H$_2$O and 15 mmol of CH$_4$N$_2$S were dissolved in 15 mL of deionized water, and formed a homogeneous solution. The mixture was then transferred into a Teflon-lined stainless-steel autoclave, heated and maintained at 140 °C for 5 h. The solid was obtained by centrifugation and washed several times with deionized water and ethanol, followed by freeze-drying treatment.

**Synthesis of CdS nanorods**. CdS nanorods were synthesized according to a literature method[70]. In all, 10 mmol Cd(NO$_3$)$_2$·4H$_2$O, 30 mmol NH$_2$CSNH$_2$, and 50 mL of ethylenediamine were added into a Teflon-lined stainless-steel autoclave, and were heated and maintained at 160 °C for 48 h. After cooling down to room temperature, a yellow precipitate was separated out and washed with absolute ethanol and deionized water several times, and then dried at vacuum oven.

**Synthesis of CdS nanowires**. CdS nanowires were synthesized according to a literature method[70]. In all, 1.6 g of Cd(NO$_3$)$_2$·4H$_2$O and 1.19 g of thiourea was added into 50 mL of ethylenediamine and stirred for 1 h. Then the mixture was transferred to a Teflon-line stainless-steel autoclave, heated at 180 °C for 72 h. The products were separated by centrifugation and washed with ethanol and deionized water, and then dried by freeze-drying treatment.

**Catalytic activity evaluation**. Photocatalytic reaction was carried out in an autoclave with top irradiation. The visible light source was a 300-W Xe lamp (PLS-SXE300/300UV, Perfect Light) equipped with a UV cutoff filter (420-780 nm). Typically, photocatalyst (10 mg) was dispersed in 20 mL solution that contained with 20 mmol substrates, 4 mL 25 wt% ammonia solution, and deaerated water. Then, the reactor was purged with nitrogen for 10 times. The photocatalytic reaction was carried out at 50 °C under 1 bar N$_2$ for certain times. After reaction, the liquid products were filtered by a PES membrane with a pore size of 0.45 µm and analyzed by high-performance liquid chromatography (HPLC, Shimazu LC-20A) equipped with Refractive Index Detector and Diode Array Detector. Ammonia was removed via evaporator or freeze dryer before analysis. Amino acids were analyzed via the derivation method with ortho-phthalaldehyde as the derivatization reagent on a Poroshell 120 EC-C8 column (4.6 × 100 mm, Agilent). The remaining lactic acid and pyruvic acid were analyzed on a Hi-plex column (8 × 300 mm, Agilent) using 10 mM H$_2$SO$_4$ aqueous solution as the mobile phase.

**Catalysts characterization**. The photocatalysts were characterized by X-ray diffraction (XRD), transmission electron microscopy (TEM), atomic force microscope (AFM) spectroscopy, electron spin resonance (ESR) spectroscopy, $N_2$ physiorption, UV-Vis spectroscopy, and photoelectrochemical tests. The details of these techniques were described in Supplementary Information.

The other experimental and computational methods were described in Supplementary Information.

## Data availability

The data that support the findings of this study are available from the corresponding authors upon a reasonable request.

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

## Acknowledgements

We thank the Tier-1 and Tier-2 projects from Singapore Ministry of Education for the financial support (R-279-000-597-114 and R-279-000-594-112). S.W. and P.H. are grateful for the financial support from the National Natural Science Foundation of China (No. 21922201 and 21872113) and the Fundamental Research Funds for the Central Universities (No. 20720190036). J.L., D.C., and J.Q. thank the National Center for International Research on Intelligent Nano-Materials and Detection Technologies in Environmental Protection. J.Q. thanks the financial support from the China Scholarship Council (CSC). We thank Xuewen Li and Hao Wu for characterizations in cyclic voltammogram tests.

## Author contributions

N.Y. and S.S. conceived the project, designed the experiments, analyzed the data, and wrote the manuscript. S.S. and J.Q. carried out the catalyst synthesis, evaluated their catalytic performances, and conducted some characterizations. G.Z. conduced photo-electrochemical characterizations. P.H. and M.J.H. performed DFT studies and analyzed the data. S.W. guided the DFT work, analyzed the data, and co-wrote the manuscript. Y.W., D.C., and J.L. analyzed the data and co-wrote the manuscript. All authors discussed the results and edited the manuscript.

## Competing interests

The authors declare no competing interests.
