## [Peer Review File · Nature Communications]

REVIEWER COMMENTS

Reviewer #1 (Remarks to the Author):

The submitted manuscript from Song et al reports the photocatalytic formation of amino acids on CdS nanosheets using biomass-derived alpha-hydroxyl acids and ammonia. Owing to importance of amino acids and the green nature of biomass-derived compounds, it is surely an interesting area to explore the synthesis of amino acids from biomass-based materials. Therefore, this study is potentially suitable for Nature Communications if the authors could successfully address the following concerns.

1. The authors mentioned a few semiconductor candidates, including TiO₂, g-C₃N₄, BiVO₄, CuS, and CdS. Although it is understood that TiO₂ is not visible light absorbing, thus it does not exhibit activities under visible light irradiation, it is not clear how those other semiconductors behave besides CdS. It seems like as long as the generated hole on a semiconductor possesses sufficient oxidizing power, that semiconductor is potentially suitable for the desirable reaction. Is that true?
2. Kinetic analysis of this work indicates the formation of alanine is half-order with respect to lactic acid and second-order to ammonia. These orders are rather odd, which needs more explanation. The authors need to provide more discussion/evidence regarding the origin of these reaction orders.
3. Once those amino acids are formed, such as alanine, will their amine groups be oxidized if the photo-irradiation time is elongated? What could be the side-products of these amination reactions? The authors are recommended to provide the cyclic voltammograms of the final amino acids and the original biomass-derived alpha-hydroxyl acids.

Reviewer #2 (Remarks to the Author):

In this manuscript the authors report a detailed experimental and theoretical investigation on the visible-light-driven amino acids production from biomass-based feedstocks over ultrathin CdS nanosheets. Several potential scenarios for the chemical synthesis of amino acids from renewable sources, as an alternative route to the current processes based on fermentation to avoid harsh reaction conditions and the use of expensive noble metal catalysts/toxic reagents, have been considered and a well-supported mechanism is proposed. The used tools and methodologies are advanced and competently used. The work is important for both fundamental and application viewpoints. Results are well presented and discussed, the bibliography is accurate and conclusions well supported by the obtained results. The manuscript can be published in the present form and I suggest to the authors to remove few typos.

Reviewer #3 (Remarks to the Author):

In this very interesting contribution Yan, Lu and Wang have proposed the first visible-light driven amino acid production from biomass derived feedstocks using CdS nanosheets as catalysts. This work is of great

importance, since it achieves (under mild conditions) the conversion of platform chemicals that can be sources from lignocellulose (lignin or cellulose) to alpha amino acids, by direct coupling with ammonia. This reaction is challenging even with simple alcohol substrates. The reaction in the paper, itself has little precedence. The authors have previously reported on this transformation using heterogeneous catalysts and ammonia, with addition of hydrogen under harsher conditions. This work now achieves, using photocatalysis, a much more efficient transformation. It is a beautiful piece of work that I recommend for publication after the minor comments have been addressed, shown below:

***Section Introduction

It would be useful if the authors could somewhat re-work this section so that the main concept and the importance of the work is better understandable for the non-expert readers.

For example, the authors say: 'Cellulose, hemicellulose and lignin are enriched with -OH groups, 17, 18 while α -amino acids bear -NH₂ group and -COOH group attached to the same carbon. The "functional group-gap" between the starting material and products require highly efficient catalytic routes to replace the -OH groups by the -COOH group¹⁹⁻²¹ and the -NH₂ group at the desired position...'

This section is too vague. Enriched with -OH groups – specify how? Replace -OH by the -COOH?

Confusing since this contribution is transforming an -OH (already in alpha position) to an -NH₂ while the -COOH is already given by another, well known transformation starting from biomass.

In this regard, The figure 1 is simple and elegant, but it would be beneficial to add slightly more details in order to familiarize the reader with how alpha-hydroxyacids are formed from lignocellulose.

-Cover the literature somewhat broader. References related to the amination of alpha hydroxyl acids or their derivatives with homogeneous or heterogeneous catalysis should be checked and cited. Recent attempts for other approaches, for the sustainable production of amino acids should be cited. For example directly from CO₂ and N₂ {e.g.

<https://pubs.rsc.org/en/content/articlelanding/2018/gc/c7gc02911j#divAbstract>]

***Results section

The section glucose to aniline would be better with more details added, perhaps the reaction scheme added with the corresponding key steps. Can the authors comment on the rationale behind using these given reaction conditions (bases, substrate concentrations...) Given that the CdS nanosheets are expected to be -OH sensitive, were there any attempts to test heterogeneous catalyst instead of common NaOH for the transformation of sugars to lactic acid? Similarly, since this system is rather different from the previous lactic acid to aniline case (more intermediates, -OH effecting) would it not be reasonable to re-test the other CdS morphologies or different photocatalysts as well which are not base sensitive? (in case other, more robust heterogeneous catalysts are not an option?)

Formatting/style:

The manuscript is well structured and well-written, only a few typos can be found, which should be corrected.

Reviewer #1 (Remarks to the Author):

The submitted manuscript from Song et al reports the photocatalytic formation of amino acids on CdS nanosheets using biomass-derived alpha-hydroxyl acids and ammonia. Owing to importance of amino acids and the green nature of biomass-derived compounds, it is surely an interesting area to explore the synthesis of amino acids from biomass-based materials. Therefore, this study is potentially suitable for Nature Communications if the authors could successfully address the following concerns.

1. The authors mentioned a few semiconductor candidates, including TiO₂, g-C₃N₄, BiVO₄, CuS, and CdS. Although it is understood that TiO₂ is not visible light absorbing, thus it does not exhibit activities under visible light irradiation, it is not clear how those other semiconductors behave besides CdS. It seems like as long as the generated hole on a semiconductor possesses sufficient oxidizing power, that semiconductor is potentially suitable for the desirable reaction. Is that true?

Response: Thank you for the comment. We calculated the reaction activation energy of two key steps in lactic acid to alanine conversion and converted them into the redox potentials. Then we plotted the conduction band and valence band positions of semiconductors tested in our study, and added these two values as reference lines (Supplementary Fig. 10 and Supplementary Table 4). It has been observed that the photocatalytic amination of lactic acid to alanine can only proceed when the conduction band (E_{VB}) position of photocatalyst is higher than the imino acid/alanine redox potential (-0.33 V) and the valence band (E_{CB}) position is lower than the lactic acid/pyruvic acid redox potential (1.05 V). This criterial matches well with experimental observation (Supplementary Fig. 1): BiVO₄ is inactive for the desired transformation since the band gap does not fall within desired range, while TiO₂, ZnO, g-C₃N₄ and various CdS materials are active for the reaction. CuS was the only exception, possibly due to its poor adsorption of organic substrates and quick recombination of the charge carriers as indicated in the literature. We have added discussions in the SI after Supplementary Fig. 10. to indicate the band gap-activity correlations.

2. Kinetic analysis of this work indicates the formation of alanine is half-order with respect to lactic acid and second-order to ammonia. These orders are rather odd, which needs more explanation. The authors need to provide more discussion/evidence regarding the origin of these reaction orders.

Response: This is an excellent comment. Considering that the break of the alpha-C-H bond is the rate-determining step, it is not unexpected to see a positive order of lactic acid. The reason why the order is not exactly 1 but at around 0.5 may be due to the partial coverage of the catalyst by lactic acid/lactate. In the extreme case of zero coverage and full lactic acid/lactate coverage, the reaction order will approach one and zero, respectively.

The second order dependence with respect to NH₃ suggests that two NH₃ molecules are involved in alpha-C-H bond breaking or lactic acid adsorption on CdS. A possible scenario would be that one NH₃ molecule interacts with the acidic carboxylic group, while a second NH₃ molecule interacts with the H in the alpha-OH group, during lactic acid adsorption and C-H bond breakage.

Prompted by the reviewer's comment, we have added the following discussions in the manuscript (page 7).

“The mechanistic assessment shown below indicates that the cleavage of C_α-H bond in lactic acid adsorbed on CdS is the kinetically-relevant step for photocatalytic amination of lactic acid. Accordingly, the measured apparent reaction order of ~0.5 on lactic acid suggests that the surface of CdS is partially covered by lactic acid at the examined reaction condition, because negligible and saturated coverages would lead to zero- and first-order kinetics, respectively. On the other hand, the second order on ammonia implies two ammonia molecules are involved in the C_α-H bond abstraction of lactic acid. We surmise that one ammonia interacts with the carboxylic H-atom in lactic acid and the other connects to the corresponding α-hydroxyl H-atom via H-bonding for stabilizing the transition state of the C_α-H cleavage step.”

3. Once those amino acids are formed, such as alanine, will their amine groups be oxidized if the photo-irradiation time is elongated? What could be the side-products of these amination reactions? The authors are recommended to provide the cyclic voltammograms of the final amino acids and the original biomass-derived alpha-hydroxyl acids.

Response: Thank you for the suggestion.

We carefully analysed the ¹H NMR spectrum of reaction mixture after photocatalytic amination of lactic acid to alanine. Apart from unconverted lactic acid and product alanine, lactamide (9.2 %), alaninamide (8.5 %), acetic acid (2.3 %) and paraldehyde (3.1%) were

detected as the main side products (Supplementary Figure 2). This also leads to the identification of reaction network which is provided as Supplementary Figure 3.

In addition, we have conducted an experiment using alanine as substrate over CdS nanosheets under visible light irradiation for elongated time (24 h). Trace amounts of pyruvic acid was detected afterwards, possibly coming from the photocatalytic oxidation of alanine. The TOC result excluded the oxidation of alanine into gas phase products such as CO₂. The ¹H NMR spectrum of alanine after 24 h exposure to photocatalytic reaction condition has been included as Supplementary Figure 4.

We have added the discussion “*Lactamide, alaninamide, acetic acid, and paraldehyde were detected as the main side products in photocatalysis amination reaction (Supplementary Fig. 2), based on which a reaction network is proposed (Supplementary Fig. 3). A control experiment using alanine as substrate demonstrated negligible further oxidation of alanine into pyruvic acid (Supplementary Fig. 4).*” into the manuscript (page 7) and related figures into the SI.

Supplementary Figure 2. ¹H NMR spectra of reaction mixture after 10 h. Reaction conditions: 10 mg CdS nanosheets, 20 mmol lactic acid, 4 mL ammonia solution (25 wt%), 16 mL deaerated water, t = 10 h, T = 50 °C, 1 bar N₂.

Supplementary Figure 3. Reaction pathway in photocatalysis amination of lactic acid to alanine.

Supplementary Figure 4. ^1H NMR spectra of alanine decomposition experiment. Reaction conditions: 2 mg CdS nanosheets, 1 mmol alanine, 2 mL D_2O , $t = 24$ h, $T = 50$ $^\circ\text{C}$, 1 bar N_2 .

Furthermore, the cyclic voltammograms of hydroxyl acids and related amino acids were conducted as suggested. The spectra have been included in the SI as Supplementary Figure 5. We have added the following discussion into the manuscript “*The cyclic voltammograms of hydroxyl acids and related amino acids were conducted (Supplementary Fig. 14). The reversible anodic and cathodic peaks were detected with hydroxyl acids, possibly coming from the redox reactions between hydroxyl acids and related keto acids. The relatively high current density of lactic acid, α -hydroxyisocaproic acid, α -hydroxy-3-methylbutyric acid, phenyllactic acid, and α -hydroxybutyric acid were observed, which may explain their faster amino acid formation rate compared to other substrates. No obvious peak was observed for amino acids except leucine, meaning most amino acids studied in this work are stable.*” (page 16).

Supplementary Figure 5. The cyclic voltammograms of difference substrates and products. **a** hydroxyl acids. **b** amino acids.

Reviewer #2 (Remarks to the Author):

In this manuscript the authors report a detailed experimental and theoretical investigation on the visible-light-driven amino acids production from biomass-based feedstocks over ultrathin CdS nanosheets. Several potential scenarios for the chemical synthesis of amino acids from renewable sources, as an alternative route to the current processes based on fermentation to avoid harsh reaction conditions and the use of expensive noble metal catalysts/toxic reagents, have been considered and a well-supported mechanism is proposed. The used tools and methodologies are advanced and competently used. The work is important for both fundamental and application viewpoints. Results are well presented and discussed, the bibliography is accurate and conclusions well supported by the obtained results. The

manuscript can be published in the present form and I suggest to the authors to remove few typos.

Response: We appreciate the very positive evaluation from this reviewer. We have checked and revised the MS to minimize errors/typos.

Reviewer #3 (Remarks to the Author):

In this very interesting contribution Yan, Lu and Wang have proposed the first visible-light driven amino acid production from biomass derived feedstocks using CdS nanosheets as catalysts. This work is of great importance, since it achieves (under mild conditions) the conversion of platform chemicals that can be sources from lignocellulose (lignin or cellulose) to alpha amino acids, by direct coupling with ammonia. This reaction is challenging even with simple alcohol substrates. The reaction in the paper, itself has little precedence. The authors have previously reported on this transformation using heterogeneous catalysts and ammonia, with addition of hydrogen under harsher conditions. This work now achieves, using photocatalysis, a much more efficient transformation. It is a beautiful piece of work that I recommend for publication after the minor comments have been addressed, shown below:

***Section Introduction

It would be useful if the authors could somewhat re-work this section so that the main concept and the importance of the work is better understandable for the non-expert readers.

For example, the authors say: ‘Cellulose, hemicellulose and lignin are enriched with -OH groups,^{17, 18} while α -amino acids bear -NH₂ group and -COOH group attached to the same carbon. The “functional group-gap” between the starting material and products require highly efficient catalytic routes to replace the -OH groups by the -COOH group¹⁹⁻²¹ and the -NH₂ group at the desired position...’

This section is too vague. Enriched with -OH groups – specify how? Replace -OH by the -COOH? Confusing since this contribution is transforming an -OH (already in alpha position) to an -NH₂ while the -COOH is already given by another, well known transformation starting from biomass.

Response: Thank you for the valuable suggestion. We have removed the confusing statement in the introduction and rewrote it as the following “*Despite various efforts to convert*

lignocellulose components into various chemicals including a number of organic acids,¹⁸⁻²² chemical transformation of biomass into amino acids is rare.” (page 3).

In this regard, The figure 1 is simple and elegant, but it would be beneficial to add slightly more details in order to familiarize the reader with how alpha-hydroxyacids are formed from lignocellulose.

Response: As suggested, we have re-drawn Scheme 1 and added related information on transformation of lignocellulose into α -hydroxyl acids. The detailed pathway for each α -hydroxyl acid is not possible to be included fully but we hope this addition provides the readers an idea of the entire picture.

-Cover the literature somewhat broader. References related to the amination of alpha hydroxyl acids or their derivatives with homogeneous or heterogeneous catalysis should be checked and cited. Recent attempts for other approaches, for the sustainable production of amino acids should be cited. For example directly from CO_2 and N_2 {e.g. <https://pubs.rsc.org/en/content/articlelanding/2018/gc/c7gc02911j#!divAbstract>}

Response: This is indeed an important point. References for the amination of α -hydroxyl acids or their derivatives have been updated and cited as [36, 37] in the manuscript. The excellent work on sustainable production of amino acids from CO_2 and N_2 (Green Chem., 2018, 20, 685-693) has also been cited as [17] in the manuscript. “For example, an exciting progress has recently been achieved on the sustainable synthesis of glycine and alanine by catalytic fixation of N_2 and CO_2 .¹⁷” (page 3).

***Results section

The section glucose to alanine would be better with more details added, perhaps the reaction scheme added with the corresponding key steps.

Response: Thank you for the valuable suggestion. We have added the reaction scheme for photocatalysis amination of glucose to alanine as Supplementary Fig. 15 in the Supporting Information.

Supplementary Figure 15. Reaction scheme for photocatalytic amination of glucose to alanine.

Can the authors comment on the rationale behind using these given reaction conditions (bases, substrate concentrations...)

Response: Thanks for the comment. The concentration of substrate was selected based on a literature report (*Green Chem.*, **2017**, 19, 76-81), which represents a common value used to convert glucose to lactic acid. We have mentioned this in the revised manuscript (page 16).

Base concentration affects the effectiveness of sugar to alanine conversion. Base is needed to promote C-C bond cleavage to generate lactic acid. On the other hand, over-high base concentration inhibits photocatalytic activity of CdS. Through screening different base concentrations, the optimum condition was identified (Supplementary Table 6).

Given that the CdS nanosheets are expected to be -OH sensitive, were there any attempts to test heterogeneous catalyst instead of common NaOH for the transformation of sugars to lactic acid?

Response: The reviewer has raised an important point. Indeed, heterogeneous catalysts are ideal options considering the potential inhibition effect of NaOH to catalyst surface. We have explored the possibility of using heterogeneous base catalyst as an alternative during paper revision. Based on a recent review (*J. Energy Chem.*, **2019**, 32, 138-151) focusing on renewable organic acid synthesis from biomass, the minimum reaction temperature to convert glucose to lactic acid is 150 °C over the best reported heterogeneous catalysts, which is too high for our photocatalytic reaction. Regrettably, it seems that there is no heterogeneous base material that can replace NaOH at this point.

Similarly, since this system is rather different from the previous lactic acid to alanine case (more intermediates, -OH effecting) would it not be reasonable to re-test the other CdS morphologies or different photocatalysts as well which are not base sensitive? (in case other, more robust heterogeneous catalysts are not an option?)

Response: Thank you for the suggestion. We tested the photocatalysis conversion of glucose to alanine over other CdS morphologies, but CdS nanosheet remain the most active one. The data has been included in Supplementary Table 6 (entries 8-11).

Entry	Photo catalyst	Alkaline	Conc./ M	Con. /%	Lactic acid/mmol	Alanine/ mmol	Alanine formation rate/ mmol g _{cat} ⁻¹ h ⁻¹
1	Nanosheets	-	-	73.6	0	0	0
2	Nanosheets	NaOH	0.20	97.5	0.48	0.021	0.26
3	Nanosheets	KOH	0.20	97.0	0.36	0.017	0.21
4	Nanosheets	LiOH	0.20	97.3	0.27	0.015	0.19
5	Nanosheets	Ba(OH) ₂	0.10	99.5	1.16	0.008	0.10
6	Nanosheets	NaOH	0.30	98.3	0.92	0.027	0.34
7	Nanosheets	NaOH	0.50	99.0	1.48	0.019	0.24
8	Nanorods	NaOH	0.30	98.1	0.94	0.023	0.11
9	Commercial	NaOH	0.30	98.6	0.90	0.022	0.08
10	Nanowires	NaOH	0.30	99.1	0.91	0.025	0.07
11	Nanospheres	NaOH	0.30	97.1	0.89	0.023	0.08

Reaction conditions: CdS 10 mg, glucose 2 mmol, 4 mL ammonia solution (25 wt%), 16 mL deaerated water, stir = 600 rpm, N₂ 1 bar, time = 8 h.

Formatting/style:

The manuscript is well structured and well-written, only a few typos can be found, which should be corrected.

Response: Thank you for the positive comments. We have checked and revised the MS to minimize errors and typos.

Finally, we thank all three reviewers for their supportive and constructive comments. Their time and encouragement are greatly appreciated.

REVIEWERS' COMMENTS:

Reviewer #1 (Remarks to the Author):

The authors tried their best to make an adequate revision, which is nearly acceptable. There are only a few minor issues that the authors are recommended to fix.

1. In Table 1, in addition to the formation rate of amino acids, it is also necessary to include the overall yields of those amino acids from their corresponding hydroxyl acids.
2. On page 7, line 133, it seems like the “zero” and “first” orders are mistakenly switched, which is in contrast to the discussion in the response letter.
3. On page 16, line 264, “entry 7” is not placed properly. It should be cited for 3-aminobutyric acid.
4. On page 17, line 304, “typical seem” should be “typically seen”.

The authors are encouraged to thoroughly proof read the whole draft to avoid any other typos/mistakes.

Reviewer #3 (Remarks to the Author):

With added references, figure changes, and additional experiments i believe that the authors have fully and convincingly addressed the points raised by the reviewers. I recommend the paper to be published as it is.

Reviewer #1 (Remarks to the Author):

The authors tried their best to make an adequate revision, which is nearly acceptable.

There are only a few minor issues that the authors are recommended to fix.

1. In Table 1, in addition to the formation rate of amino acids, it is also necessary to include the overall yields of those amino acids from their corresponding hydroxyl acids.

Response: Thank you for the suggestion. We have added the overall yield in the Table 1.

2. On page 7, line 133, it seems like the “zero” and “first” orders are mistakenly switched, which is in contrast to the discussion in the response letter.

Response: We are sorry for the mistake. We have revised the description in the manuscript.

3. On page 16, line 264, “entry 7” is not placed properly. It should be cited for 3-aminobutyric acid.

Response: We have placed “entry 7” in the right position in the manuscript.

4. On page 17, line 304, “typical seem” should be “typically seen”.

Response: Thank you for the comments. We have revised the mistake.

The authors are encouraged to thoroughly proof read the whole draft to avoid any other typos/mistakes.

Response: We have checked and revised the MS to minimize errors/typos. Thanks for the comments and efforts from this reviewer.

Reviewer #3 (Remarks to the Author):

With added references, figure changes, and additional experiments i believe that the authors have fully and convincingly addressed the points raised by the reviewers. I recommend the paper to be published as it is.

Response: We appreciate the very positive evaluation from this reviewer.